# Macrophages Treated with VEGF and PDGF Exert Paracrine Effects on Olfactory Ensheathing Cell Function

**DOI:** 10.3390/cells11152408

**Published:** 2022-08-04

**Authors:** Souptik Basu, Indra N. Choudhury, Jia Yu Peppermint Lee, Anu Chacko, Jenny A. K. Ekberg, James A. St John

**Affiliations:** 1Clem Jones Centre for Neurobiology and Stem Cell Research, Nathan Campus, Griffith University, Nathan, QLD 4222, Australia; 2Menzies Health Institute Queensland, Southport Campus, Griffith University, Southport, QLD 4222, Australia; 3Griffith Institute for Drug Discovery, Nathan Campus, Griffith University, Nathan, QLD 4111, Australia

**Keywords:** growth factor, spinal cord injury, glia, phagocytosis, myelin, NF-κB

## Abstract

Glial cell transplantation using olfactory ensheathing cells (OECs) holds a promising approach for treating spinal cord injury (SCI). However, integration of OECs into the hostile acute secondary injury site requires interaction and response to macrophages. Immunomodulation of macrophages to reduce their impact on OECs may improve the functionality of OECs. Vascular endothelial growth factor (VEGF) and platelet-derived growth factor (PDGF), known for their immunomodulatory and neuroprotective functions, have provided improved outcomes in SCI animal models. Thus, VEGF and PDGF modulation of the SCI microenvironment may be beneficial for OEC transplantation. In this in vitro study, the effect of VEGF and PDGF on macrophages in an inflammatory condition was tested. Combined VEGF + PDGF reduced translocation nuclear factor kappa B p65 in macrophages without altering pro-inflammatory cytokines. Further, the ability of OECs to phagocytose myelin debris was assessed using macrophage-conditioned medium. Conditioned medium from macrophages incubated with PDGF and combined VEGF + PDGF in inflammatory conditions promoted phagocytosis by OECs. The growth factor treated conditioned media also modulated the expression of genes associated with nerve repair and myelin expression in OECs. Overall, these results suggest that the use of growth factors together with OEC transplantation may be beneficial in SCI therapy.

## 1. Introduction

The transplantation of olfactory ensheathing cells (OECs) to repair spinal cord injury (SCI) has gained considerable interest over the last two decades [1]. The unique ability of OECs to promote continuous olfactory neuron growth throughout life has been translated into novel therapies [2]. When transplanted into SCI, OECs have been observed to interact with glial scar tissue, promote neovascularisation, clear inhibitory axonal debris and help in axonal sprouting and remyelination, with the overall outcome of promoting functional recovery [3,4]. However, the biggest pitfall in the transplantation of OECs has been the difficulty of their integration, survival and functional efficacy at the injury site [5].

SCI follows a transient primary and a progressive secondary phase. The latter is characterised by a multitude of cellular and biochemical reactions [6]. The secondary phase begins with the recruitment and activation of the neutrophils, which is then followed by a gradual influx of circulating monocytes, slowly differentiating into macrophages [7]. The temporal profile of macrophage settlement at the injury site is crucial as it governs the timing of an effective therapeutic intervention. In rodent SCI models, macrophages start increasing around three days and have a bimodal pattern of distribution, reaching two peaks at day 7 and day 60 [8]. After populating the injury site, macrophages play vital roles spanning from a negative, deleterious effect of the M1 (pro-inflammatory) subset of macrophages to the positive, beneficial effect of the M2 (anti-inflammatory) macrophages. While the former causes neurotoxicity, the latter is concerned with early recovery and healing [9,10]. In SCI, unlike normal wound healing, M1 macrophages predominate over M2, leading to a hostile microenvironment that is unfavourable for cellular regeneration [9,11]. Thus, immunomodulation by targeting this population becomes a key intervention in developing a potential therapy for SCI [12,13]. Several studies have been recently conducted that focus on this aspect and improve the healing process after SCI [14,15,16].

Vascular endothelial growth factor (VEGF) is a key angiogenic factor which has also been linked with neurotrophic activity [17,18]. For macrophages, VEGF has been reported to show significant modulatory activities and plays a vital role in its recruitment, which thereby promotes neovascularization in inflammation [19,20]. In SCI, VEGF can reduce inflammation through autophagy, which leads to the promotion of earlier recovery [21,22].

Platelet derived growth factor (PDGF) is another mitogen similar to VEGF. It is crucial for cells originating from mesenchyme, namely, fibroblasts, smooth muscle cells and oligodendrocytes [23]. Its chief role is ensuring recruitment of the smooth muscle cells and stabilising newly sprouted micro-vessels [24]. Additionally, PDGF is also important in macrophage recruitment for maturation of these blood vessels [25]. Additionally, PDGF has been linked with neuroprotective activity against oxidative stress and is implicated in a beneficiary role in ischemic strokes in humans [26,27]. In SCI, PDGF has been administered successfully in rats, leading to increased survival of oligodendrocyte precursor cells which provide protection to motor neurons [28]. Owing to their respective positive roles in SCI, the combined application of VEGF and PDGF has been tested, with results demonstrating a reduction of secondary injury area and gliosis and a beneficial modulation of macrophages [29]. However, this first study did not facilitate motor recovery, likely due to the short-term application of the growth factors. Hence, a successive study examined the sustained release of combined VEGF + PDGF over a week, which led to improved behavioural and histological outcomes [30]. While these studies demonstrated efficacy, they lacked an understanding of inter-cellular interaction complexities and subsequent perturbations in the SCI microenvironment. Thus, in vitro dissection of such an interaction would help us optimise our therapeutic approach.

While immunomodulation can exert benefits, it is unlikely to lead to a complete recovery on its own. Therefore, additional therapeutic strategies that target neural regeneration and rescue of neural functions become pivotal strategies. This could be achieved by cell transplantation, using cells such as OECs, which can secrete and supply neurotrophic factors and provide support and guidance for sprouting axons [31]. However, the failure of OECs to survive and integrate can reduce their therapeutic efficacy [5]. Macrophages are a dominant cell in the acute stage of SCI and their interaction with the transplanted OECs becomes very important as it may influence the activity of the OECs. Previously, our group had shown that OECs express macrophage migration inhibitory factors (MIF) and are responsible for reduction in the recruitment of macrophages in vitro and aid in the phagocytosis of axonal debris by OECs [32]. Thus, macrophage–OEC interactions are clearly modulated, at least to some degree.

In our present in vitro study, we aim to determine the paracrine effects of VEGF and PDGF-treated macrophages on OECs in the presence or absence of an inflammatory setting. Our specific aims are to ascertain whether VEGF and PDGF can initiate an inflammatory reaction in macrophages and how it changes their subsequent inflammatory cytokine secretion profile. We treated the macrophages for 3 h, 24 h and 48 h with variable conditions. The rationale behind choosing these time points is related to the fluctuating Toll-like receptor-4 (TLR-4) activation in the macrophages linked with variable inflammatory signalling and has been used in previous studies [33,34,35]. Thus, we hypothesise that conditioned media from these variable activated states of macrophages can modulate the phagocytic activity and gene expression of OECs. Overall, this in vitro study aims to reveal the complex growth factor versus cell interactions that potentially occur during the application of a combined VEGF/PDGF plus OEC transplantation therapy for SCI repair.

## 2. Materials and Methods

### 2.1. Ethics

All procedures were approved by the University Animal Ethics Committee (MSC/13/18) and by the University Biosafety Committee (NLRD/003/2020_Var2) of Griffith University under the guidelines of the National Health and Medical Research Council of Australia and in accordance with the Australian Code for the Care and Use of Animals for Scientific Purposes (8th Edition, 2013), and in accordance with the Australian Commonwealth Office of the Gene Technology Regulator.

### 2.2. Reagents

A pro-inflammatory environment was created using a combination of interferon-γ (IFN-γ) (1 ng/mL, Abcam, Cambridge, UK) and lipopolysaccharide (LPS) from *E. coli* O111:B4 (100 ng/mL, Sigma, St. Louis, MI, USA), as suggested previously [36]. For evaluating the effect of growth factors, we used recombinant human VEGF-165 protein (Gibco, Waltham, MA, USA) and recombinant mouse PDGF-BB protein (Gibco) [36].

### 2.3. Primary Peritoneal Macrophage Culture

A primary peritoneal macrophage culture was prepared following a previously established protocol [37]. Briefly, C57Bl/6 mice were used at eight weeks for collection of the cells. A 3% thioglycollate medium brewer (BD biosciences, Franklin Lakes, NJ, USA) solution was prepared and sterilised prior to use. On day 1, it was injected intraperitoneally (1 mL/mouse) under anaesthesia using a 26G needle. On day 7, the cells were harvested after euthanising the mice and lavaging the peritoneum with 5 mL of ice cold 1X phosphate buffered saline (PBS) (Gibco). The collected cells were centrifuged at 500× *g* for 5 min and plated out in a 6-well plate in a complete culture medium comprising of Dulbecco’s Modified Eagle Medium (DMEM) (Gibco), 10% fetal bovine serum, gentamycin (50 μg/mL, Gibco) and L-glutamine (200 μM, Gibco). After 18 h, the media was changed completely, to remove any debris or traces of floating monocytes. After this, half the media was changed once before using it on the third day. The purity of cells was assessed by staining with F4/80 antibody (1:200, Abcam). Based on this protocol, we achieved 89–93% pure macrophages.

### 2.4. Primary OEC Culture

Primary OEC culture was performed following previously established protocols [38,39,40]. Briefly, S100β-DsRed transgenic mouse pups, at P7–P9, were decapitated in sterile conditions. After this, they were hemi-sectioned by sagittal section and placed under a stereo-dissection microscope (Olympus, Tokyo, Japan). Only the anterior one-third portion of the olfactory bulb area (nerve fibre layer) with bright red fluorescence was extracted, chopped into fine pieces and dispensed in 1% G5 (Gibco) supplemented media (DMEM + 10% FBS) in a Matrigel-coated 24-well plate. The explants were left to attach for 30 min at room temperature before being transferred to an incubator at 37 °C and 5% CO_2_. After 24–48 h, complete media removal was done to clear debris, and then every alternate day, half the media was changed until confluency was reached. Upon reaching confluency, the cells were detached using Tryp-LE express (1X, Gibco), followed by centrifugation, and were plated in T25 culture before being used for experiment. This extraction process yielded 70–80% of p75NTR/DsRed positive glial cells as established by our methods mentioned previously.

### 2.5. NF-κB p65 Translocation Assay

The NF-κB p65 translocation assay was performed by seeding primary peritoneal macrophages at 4000 cells/well in a 384-well plate (Greiner Bio-One, Kremsmünster, Austria). They were kept in complete media (DMEM + 10% FBS and gentamycin) for overnight incubation. The next day, the media was replaced with serum-free media containing the following conditions: inflammatory-only (IFN-γ + LPS), growth factors (VEGF at 50 ng/mL, PDGF at 50 ng/mL, combined VEGF + PDGF at 50 ng/mL each), inflammatory with growth factors (IFN-γ + LPS + VEGF, IFN-γ + LPS + PDGF, IFN-γ + LPS + VEGF + PDGF) and an untreated control. They were fixed using 4% paraformaldehyde (PFA) at 3, 24 and 48 h. Further, they were stained using NF-κB p65 (1:500, Abcam), F4/80 (Abcam) and Hoechst. Images were acquired using a Nikon Ti2 widefield microscope (Tokyo, Japan) at 20× objective. They were processed using CellProfiler 3.1.9 software (Broad Institute, MIT, Cambridge, MA, USA) to assess nuclear: cytoplasm NF-κB p65 ratio.

### 2.6. Enzyme-Linked Immunosorbent Assay (ELISA) of Cytokines and Preparation of Macrophage Conditioned Medium

Macrophages were seeded at 1 × 10^6^ cells/well in a 6-well plate and grown to 80% confluency, after which the complete media was replaced with serum-free activated media containing the following conditions for baseline secretion of cytokines: (i) no treatment (control) or the growth factors (ii) VEGF, (iii) PDGF, (iv) VEGF + PDGF. For cytokines in inflammatory conditions, the following groups were used: (i) IFN-γ + LPS, (ii) IFN-γ + LPS + VEGF, (iii) IFN-γ + LPS + PDGF and (iv) IFN-γ + LPS + VEGF + PDGF. For extraction of cytokine, cell culture supernatant was collected after 3, 24 and 48 h treatment and then centrifuged at 1500 rpm for 10 min at 4 °C. Supernatant samples were divided into two parts and stored at −80 °C. One part of it was used as macrophage conditioned medium (MCM), and the other was used for ELISA.

For ELISA, the supernatants were used as neat, 1 in 10 or 1 in 100 dilutions with ELISPOT diluent, and readings were taken that were within the detectable range of the kits. Two different pro-inflammatory cytokine kits were used for our sandwich ELISA assays: IL-6 (Invitrogen, 88-7064-86, Waltham, MA, USA) and TNF-α (Invitrogen, 88-7324-86). The entire assay was performed as per the manufacturer’s protocol. We incubated the samples overnight at 4 °C for more sensitivity as per recommendation. Plate readings were taken using absorbance values at 450 nm and 570 nm under the POLARstar Omega plate reader (BMG Labtech, Victoria, Australia). Analysis of the data was done by first subtracting the readings of 570 nm from 450 nm.

### 2.7. Myelin Debris Preparation

Myelin was prepared following a previously established procedure [38]. A total of 10–12, S100β-DsRed transgenic mice (12 months old) were euthanised. Their entire brains were removed and homogenised in 0.32 M sucrose solution (diluted from 1 M sucrose solution in Tris-Cl buffer) using TissueLyser II (Qiagen, Hilden, Germany). The homogenate was then layered on top of a 0.83 M sucrose solution in an ultracentrifuge tube and centrifuged at 100,000× *g* for 45 min at 4 °C. The interface between the two layers was carefully pipetted out, containing myelin debris. This was then further homogenised and resuspended in Tris-Cl buffer. This centrifugation step was repeated twice, and the final myelin pellet was washed in PBS and centrifuged at 22,000× *g* for 10 min at 4 °C. The final pellet was weighed and stored at a concentration of 50 mg/mL at −80 °C, until further use.

### 2.8. Phagocytosis of Myelin Debris

OECs were seeded at 4000 cells/well and incubated overnight in supplemented media. Myelin debris was prepared as above and labelled with pHrodo Green STP Ester dye (pHrodo STP; Thermofisher, Waltham, MA, USA), as per the manufacturer’s instructions. The myelin stock solution was thawed and homogenised with an insulin syringe 5–10 times before being centrifuged at 14,000× *g* for 10 min at 4 °C. The supernatant was removed and then the pellet was resuspended in PBS (final concentration of 5 mg/mL). A labelling buffer containing 0.1 M NaHCO_3_ (pH 8.3) was prepared and filtered using a 0.22 µm surfactant-free cellulose acetate membrane filter unit (Merck, MA, USA). The pHrodo STP stock (2 mg/mL; 2 mM) was then diluted to 12.5 µM in the labelling buffer. The diluted myelin stock was centrifuged, resuspended in 1 mL of pHrodo STP-labelling buffer and put on a shaker for 1–2 h. Further, the debris was washed with PBS 4–5 times. Finally, the pellet was weighed and resuspended to a 1 mg/mL concentration. For the phagocytosis assay, live cell imaging was done using the Incucyte live cell imaging system (Sartorius, Göttingen, Germany). As the OECs express DsRed protein and myelin debris was labelled with pHrodo green, images were acquired every 30 min, with an exposure of 150 ms in the red and green channels, respectively. Analysis of phagocytosing cell was performed using CellProfiler 4.2.1 software (Broad Institute, MIT, Cambridge, MA, USA).

### 2.9. Real Time Quantitative Polymerase Chain Reaction (qPCR)

For qPCR assays, 4 × 10^5^ cells/well were seeded in a 24-well microplate (Greiner Bio-One) in complete media, overnight in 5% CO_2_ and at 37 °C. The following day, the media was substituted with serum-free media and the conditions were included as mentioned previously for the ELISA assay. For direct effects of growth factors, the cells were directly treated with growth factors (VEGF, PDGF and combined VEGF + PDGF) in the presence and absence of inflammatory mediator (IFN-γ + LPS). After 24 h of incubation, the media was removed, washed with 1× PBS and the cells were lysed using RNA lysis buffer (PureLink RNA minikit, Invitrogen) with 1% mercaptoethanol (Gibco). The samples were stored at −80 °C until further processing. On the day of the assay, the samples were homogenized, washed with 70% ethanol and centrifuged at 12,000× *g* for 15 s. After a further few washes with buffers (Purelink RNA minikit, Invitrogen), RNA was eluted using 30 µL of UltraPure DNase/RNase-Free distilled water (Invitrogen). The quality and amount of RNA were measured using a NanoDrop 1000. cDNA was synthesised using SuperScript IV VILO Master Mix with ezDNase Enzyme (Invitrogen) using the manufacturer’s protocol. cDNA samples were diluted to 10 ng/µL and used for qPCR reactions in a Quantstudio 6 flex lightcycler (Thermofisher). For each 10 µL RT-qPCR reaction, we used 5 µL of PowerUp SYBR green Master mix (2X, Thermofisher), 500 nM of Forward and Reverse primers each (Appendix A), 1 µL of cDNA and rest water. The melting temperatures of all primers were above 60 °C. PCR reaction settings were as follows: (a) Uracil DNA-glycosylase activation at 50 °C for 2 min, (b) Dual-Lock DNA polymerase at 95 °C for 2 min, (c) denaturation at 95 °C for 15 sec with annealing and extension at 60 °C for 1 min repeated for 40 cycles. A melt curve was generated to detect any primer-dimer formation.

### 2.10. Statistical Tests and Graphs

For statistical evaluations and graphical representations, GraphPad Prism 9 (San Diego, CA, USA) was used. All data were expressed as the mean ± standard error of the mean (SEM) and statistical significances were represented in each figure. Comparisons of two groups were done by unpaired *t*-test (two-tailed) with Welche’s correction or the Kolmogorov–Smirnov test, as where appropriate. For multiple groups, either one-way or two-way analysis of variance (ANOVA) was used, as appropriate, and was followed by a post hoc test comprising of Dunn’s or Fisher’s LSD test, respectively.

## 3. Results

### 3.1. NF-κB p65 Translocation in Macrophages Is Increased by PDGF in Inflammation

NF-κB is a large family of proteins responsible for triggering inflammatory genes by translocating from cytoplasm to nucleus through canonical and non-canonical pathways [41,42]. To help determine the duration of growth factor therapy, it is imperative to consider the dynamics of NF-κB translocation from cytoplasm to nucleus in macrophages, which could ultimately modulate the functionality of OECs.

The translocation of NF-κB was determined by immunostaining with an antibody against NF-κB p65 subunit and calculating the nuclear-cytoplasm (N:C) intensity ratio. A representative image of macrophages treated for 48 h in IFN-γ + LPS with and without VEGF/PDGF is shown for comparison (Figure 1A–D). Macrophages were first cultured in untreated control medium or medium with IFN-γ + LPS for 3 h (Figure 1E). There was significant translocation of NF-κB in the inflammatory medium (medium with IFN-γ + LPS) (Figure 1E). To evaluate the basal effects of growth factors, macrophages were then exposed to VEGF and PDGF as single or combinational therapies. Both VEGF and PDGF reduced translocation of NF-κB, while the combined VEGF + PDGF had no significant changes (Figure 1F). To mimic an injured site, we next treated the macrophages with the different growth factor groups in the presence of inflammatory medium. No significant changes in translocation were observed following 3 h exposure in between the groups (Figure 1G). Thus, where VEGF and PDGF by itself can reduce NF-κB translocation, in the presence of inflammatory medium it had no effect when treated for 3 h.

Upon further increasing the length of treatment to 24 h, IFN-γ + LPS-treated macrophages still showed increased translocation (Figure 1H). Interestingly, combined VEGF + PDGF significantly reduced translocation compared to PDGF and untreated control (Figure 1I). However, 24 h-treated inflammatory media with and without VEGF/PDGF did not significantly alter translocation, similar to the 3 h treatment (Figure 1J). Therefore, after 24 h exposure, where combined VEGF + PDGF by itself can reduce NF-κB translocation, in the presence of inflammatory signal there is no change.

Prolonged exposure of macrophages to inflammatory media for 48 h, showed no significant difference in N:C ratio (Figure 1K). However, both PDGF and combined VEGF + PDGF increased NF-κB p65 translocation in comparison to VEGF and untreated control (Figure 1L). When further challenged in the presence of IFN-γ + LPS, all the combinations of VEGF and PDGF increased NF-κB translocation. Comparatively, PDGF in inflammatory media considerably increased translocation compared to others (Figure 1M).

Thus, we concluded that IFN-γ + LPS stimulates NF-κB translocation in macrophages for up to 24 h. At baseline conditions, while VEGF and PDGF, individually, can reduce translocation at 3 h treatment, their combination was effective at 24 h treatment. However, till 24 h of inflammatory exposure, no changes between the growth factors could be determined. On prolonged exposure to 48 h, PDGF by itself and in the inflammatory system can be detrimental by increasing NF-κB translocation and triggering inflammatory gene transcription.

### 3.2. PDGF Reduces Pro-Inflammatory Cytokine TNF-α in Inflammatory Media

In the previous section, we observed that NF-κB translocation in macrophages is dependent on the duration of exposure and type of growth factor. Hence, our next aim was to determine the cytokine levels of inflammatory proteins [42]. We assessed the gene expression and cytokine secretion of two known pro-inflammatory factors: tumour necrosis factor alpha (TNF-α) [43] and interleukin-6 (IL-6) [44].

When the macrophages were exposed to inflammatory media (IFN-γ + LPS), a significant up-regulation of TNF-α gene expression across all time points was detected (Figure 2A). None of the growth factors by themselves led to any change in expression (Figure 2B). In fact, when the macrophages were challenged with growth factors in inflammatory media (IFN-γ + LPS), there was no difference in TNF-α gene expression between the groups (Figure 2C). Thus, the inflammatory media stimulates TNF-α gene expression, but the growth factors do little to influence TNF-α gene expression in these conditions.

Next, we measured the corresponding TNF-α cytokine level from the supernatant. We observed that inflammatory challenged (IFN-γ + LPS) macrophages expressed significant amounts of TNF-α across all time points compared to the control. For untreated controls, there was no detectable levels of TNF-α at 3 and 24 h, unlike at 48 h (Figure 2D). This trend was similar to our previously obtained PCR data (Figure 2A). We next measured the baseline secretion of TNF-α from VEGF, PDGF and combined VEGF + PDGF-treated macrophages. At 3 and 24 h, there were no detectable levels of TNF-α, however, at 48 h VEGF and PDGF siginifcantly reduced TNF-α levels (Figure 2E). When we next challenged the macrophages with growth factors in inflammatory media, all of the time points produced detectable levels of TNF-α, however, PDGF, at 24 h, significantly suppressed the TNF-α production when compared to inflammatory-only media (Figure 2F).

We next assessed the expression of IL-6 in the macrophages when exposed to different conditions. We observed that IL-6 expression, unlike TNF-α, only showed significant up-regulation at a very late stage (48 h) (Figure 2G). When we exposed the macrophages to VEGF, PDGF and combined VEGF + PDGF, no significant changes in expression of IL-6 were detected at 3 h and 48 h. However, at 24 h, only PDGF showed up-regulation compared to combined VEGF + PDGF (Figure 2H). Further, when we challenged the macrophages with growth factors in inflammatory conditions, no significant difference between the groups was observed (Figure 2I). For IL-6 cytokine production, inflammatory-only media produced significant levels of IL-6 for 24 h and 48 h timepoints. Similar to TNF-α, untreated control showed detectable levels of IL-6 only at 48 h (Figure 2J). When we assessed the effect of growth factors on IL-6 levels, they did not produce any detectable levels across all the time points (Figure 2K). However, when we challenged with inflammatory media and growth factors, such as TNF-α, all of them produced detectable levels of IL-6, but there was no significant difference between the groups (Figure 2L).

Hence, inflammatory media stimulated TNF-α, at both transcriptional and translational levels, from 3–48 h. With an increase in time (48 h), unstimulated macrophages can lead to higher levels of TNF-α, but this would be suppressed by both VEGF and PDGF. Under inflammatory conditions, PDGF provides protection by reducing TNF-α production at 24 h. The changes in IL-6 expression and cytokine production were evident when macrophages were treated for 24–48 h, showing the difference in dynamics with TNF-α. However, no difference existed between the inflammatory growth factor groups for IL-6.

### 3.3. Growth Factor Treated Macrophage Conditioned Medium Influences Phagocytic Activity of OECs

The accumulation of myelin debris in the injury site is an inevitable event in SCI, and it triggers neuroimmune interactions [45]. Importantly, the myelin lipids are inhibitory to axonal growth and branching [46]. Thus, to improve regeneration, therapeutic approaches that enhance the clearance of the myelin components are critical [47].

Glial cells, such as Schwann cells (SCs) and astrocytes, have been used in the past for myelin debris clearance [48,49]. Additionally, we have previously shown that OECs, similar to SCs, are able to quickly engulf myelin debris in vitro [40]. As the survival and function of OECs that are transplanted into the inflammatory injury site are influenced by the resident cells, including macrophages [5], we hypothesised that the responses can be modulated by the application of the growth factors VEGF and PDGF to macrophages. Hence, our next aim was to determine how macrophages that have been treated with the growth factors, in the presence and absence of inflammatory media, can regulate the phagocytic ability of OECs, a key function in SCI repair.

We tagged the myelin debris with pHrodo dye which fluoresces green as soon as the debris is internalised into the acidic environment of lysosomes (pH~4.0). Thus, this approach provides information about the rate of digestion of the debris by OECs. An image-based assay was used to track the debris uptake by the DsRed-OECs of green, fluorescent myelin debris every half an hour using the live cell imaging system (Incucyte) (Figure 3A–H). Phagocytosis was determined by counting the percentage of DsRed objects (OECs) overlapping with a green, fluorescent object (myelin debris within a phagolysosome) to the total number of DsRed objects (OECs) present in that field of view (FOV). Since glial cells, similar to OECs, have a slower profile in debris uptake than professional phagocytes (macrophages) [40], an area under curve (AUC) was constructed using the percentage of phagocytic DsRed objects, to determine the efficacy of phagocytosis over 24 h.

We first measured the phagocytic ability of OECs when directly exposed to the growth factors with/without the inflammatory media. There was no significant difference in phagocytosis (No MCM; Figure 3I). However, when OECs were grown in conditioned media from macrophages (MCM), significant differences were observed across 3, 24 and 48 h (MCM; Figure 3I). For 3 h MCM, inflammatory media had a lower efficacy which improved in 24 h MCM but lowered again in 48 h MCM. Thus, we concluded that while OECs, directly, are unable to engulf myelin debris efficiently, through MCM they can improve engulfment when macrophages are exposed to 24 h of inflammatory conditions.

Next, we assessed the role of growth factors on OEC phagocytosis. PDGF produced the most significant effect over control, VEGF, and combined VEGF + PDGF (No MCM; Figure 3J). Growth factor treated MCM (3 and 24 h) also produced significant debris uptake in OECs. PDGF and combined VEGF + PDGF produced the most significant changes in OECs treated with 3 h MCM. Additionally, for 24 h MCM treated OECs, VEGF improved phagocytosis along with PDGF and combined VEGF + PDGF. No differences could be determined for 48 h MCM (MCM; Figure 3J). This led us to conclude that PDGF could improve phagocytosis of OECs both by itself as well as through MCM, while combined VEGF + PDGF and VEGF could only produce such a change through 3 h and 24 h MCM.

Finally, we assessed the role of growth factors in inflammatory media to influence OEC phagocytosis. Unlike previously, PDGF could not produce efficient phagocytosis. In fact, it reduced the uptake along with VEGF when compared to inflammatory only media. In contrast, combined VEGF + PDGF, could maintain phagocytosis like inflammatory media (No MCM; Figure 3K). Our MCM assay showed that 3 h MCM produced less efficient phagocytosis. VEGF was the least efficient amongst all the conditions (3 h; MCM; Figure 3K). As for 24 h MCM, combined VEGF + PDGF (Appendix A) produced the least efficient phagocytosis compared to all conditions under inflammatory media (24 h; MCM; Figure 3K) (Appendix A). When treated with 48 h MCM, all growth factor treated OECs in inflammatory media produced efficient phagocytosis, of which PDGF was the most effective (48 h; MCM; Figure 3K).

In conclusion, OEC phagocytosis of myelin debris is greatly influenced by macrophages, which depends on the type and duration of treatment that they have been exposed to. Unconditioned media (no MCM) do produce some functional changes in OECs, but not as efficiently as MCM. PDGF produced the most significant phagocytosis when used by itself on OECs or when administered on macrophages with (3 and 24 h) and without (48 h) inflammatory media.

### 3.4. Growth Factor Treated Macrophage Conditioned Media Influences Nerve Repair and Myelin Expression Associated Genes in OECs

OECs are known for their myelination and nerve repairing properties [50,51]. The influence of macrophages on expression of nerve repairing- (*Jun*, *Ngfr*, *Bdnf*, *Sox2, Gdnf*) and myelin expression-associated genes (*Egr2*, *Sox10*, *Mpz*, *Pou3f1*, *Srebf1*) would, therefore, be critical in transplantation, especially at the injury site. Moreover, with the addition of VEGF and PDGF to macrophages, further modulation of the expression of these genes in OECs can be speculated. Thus, in our next assay we compared the expression pattern of these gene groups in OECs under the presence and absence of conditioned media from growth factor-treated macrophages. All the genes were normalised to untreated control in unconditioned media.

We first assessed the effect of inflammatory media on nerve repair associated genes in OEC. We found that inflammatory media upregulated all the nerve repair associated genes compared to control in unconditioned media (Figure 4A). However, conditioned media showed a variable response. For the 3 h MCM, inflammatory media upregulated both *Jun* and *Ngfr* and downregulated *Sox2* and *Gdnf* (Figure 4B). No change in expression was noted in the 24 h MCM (Figure 4C). Similarly, 48 h inflammatory MCM only downregulated *Jun* without altering other genes (Figure 4D). Next, we assessed the effect of growth factors in non-inflammatory media. PDGF upregulated *Jun*, *Ngfr* and *Gdnf* while VEGF only upregulated *Bdnf* and downregulated *Gdnf* compared to control (Figure 4E). Assessment of MCM containing growth factors showed upregulation of most of the genes: VEGF (*Jun*, *Ngfr*), PDGF (*Jun*, *Ngfr*, *Bdnf*), VEGF + PDGF (*Ngfr*, *Sox2)* (Figure 4F). However, no changes were observable for 24 h and 48 h MCM (Figure 4G,H). As in SCI, inflammation plays a major role in determining the outcome of a therapy [52], we next compared the nerve repair-associated gene expressions in OECs treated with growth factors in inflammatory media. We wanted to observe if growth factors provided an added benefit by upregulating genes in inflammatory media. Therefore, we compared the expressions with inflammatory only media. Our unconditioned media showed that VEGF downregulated the gene, *Sox2,* but did not alter others (Figure 4I). However, when we added conditioned media, major changes were seen in 3 h MCM (Figure 4J) compared to 24 h (Figure 4K) and 48 h MCM (Figure 4L). Both *Jun* and *Ngfr* were downregulated when compared to inflammatory only media for all growth factor combinations in 3 h MCM (Figure 4J). While VEGF only upregulated *Bdnf*, both PDGF and combined VEGF + PDGF upregulated *Sox2*. In addition, combined VEGF + PDGF also upregulated *Gdnf* (Figure 4J). Thus, combined VEGF + PDGF in inflammatory conditions could be speculated to provide a beneficial effect by upregulating more nerve repair associated genes. Further assessment of other MCMs did not show major changes except for 48 h MCM where both VEGF and PDGF upregulated *Jun* when compared to inflammatory only media (Figure 4L).

We next assessed the myelin expression associated genes in OECs. We first wanted to see the effect of inflammatory media, with or without conditioned media, on the gene expression. Our unconditioned media showed that only three genes (*Egr2*, *Sox10* and *Mpz*) were upregulated by inflammatory media while *Pou3f1* was downregulated compared to the control (Figure 5A). Next, when we treated them with 3 h MCM, both *Sox10* and *Pou3f1* were significantly downregulated compared to control (Figure 5B). However, both 24 h and 48 h MCM showed no alteration in expression (Figure 5C and Figure 5D, respectively). We then assessed the effects of growth factors. Unconditioned media downregulated most of the genes. PDGF downregulated all genes except *Sox10*. However, VEGF and combined VEGF + PDGF downregulated only *Egr2* and *Srebf1,* respectively (Figure 5E). However, MCM containing growth factors showed that PDGF first upregulated *Mpz* with 3 h MCM (Figure 5F), followed by downregulation for 24 h MCM (Figure 5G), while again upregulating for 48 h MCM (Figure 5H). Interestingly, for *Pou3f1,* both PDGF and combined VEGF + PDGF first downregulated for 3 h MCM (Figure 5F), followed by upregulation for 24 h MCM (Figure 5G). Additionally, combined VEGF + PDGF could only upregulate *Srebf1* for 3 h MCM (Figure 5F). Similar to nerve repair associated genes, we next compared the growth factors in inflammatory media with inflammatory media only for their role in myelin expression in OECs. In unconditioned inflammatory media, only *Egr2* was significantly downregulated by VEGF (Figure 5I). Our conditioned media comparisons showed that in 3 h MCM while combined VEGF + PDGF upregulated *Pou3f1,* VEGF downregulated the expression compared to inflammatory only media (Figure 5J). Interestingly, *Mpz* was downregulated by VEGF and combined VEGF + PDGF for 24 h MCM (Figure 5K) but was upregulated by the latter for 48 h MCM (Figure 5L). However, *Sox10* was downregulated by PDGF for 24 h MCM (Figure 5K) and VEGF for 48 h MCM (Figure 5L).

Thus, both unconditioned and conditioned media greatly influenced OEC gene expression for both groups. Unconditioned media with inflammatory mediators could increase the nerve repair genes’ expression (Figure 4A) better than myelination (Figure 5A), while most of the growth factors (either VEGF or PDGF) promoted nerve repair genes (Figure 4E) over myelination genes (Figure 5E). However, growth factors could not provide enhanced effects on inflammation for either group of genes (Figure 4I and Figure 5I). For conditioned media, only 3 h MCM could influence variable expression of nerve repair associated genes in OECs (Figure 4B,F), compared to 24 h (Figure 4C,G) and 48 h MCM (Figure 4D,H). The effect of 3 h inflammatory MCM was equivocal (downregulated *Sox2*, *Gdnf*; upregulated *Jun*, *Ngfr*) (Figure 4B). However, combined VEGF + PDGF, proved to be the most beneficial as it upregulated both under inflammatory and non-inflammatory conditions more nerve repair genes than either VEGF or PDGF alone (Figure 4F,J). For myelination genes, unlike unconditioned media, inflammatory media for 3 h MCM showed no upregulation of the genes (Figure 5B), whereas 24 h MCM (Figure 5C) and 48 h (Figure 5D) did not produce any effect at all. For growth factors, both PDGF and combined VEGF + PDGF, under both inflammatory (Figure 5J–L) and non-inflammatory conditions (Figure 5F–H), showed equivalent response with defined patterns in *Pou3f1* and *Mpz* gene expressions (as described above), across all three MCMs. Hence, combined VEGF + PDGF in macrophage conditioned media can provide beneficial effects in OEC transplantation.

## 4. Discussion

This study has assessed how macrophage conditioned medium affects the phagocytic ability and gene expression pattern in OECs and the influence of the growth factors VEGF and PDGF. Macrophages exposed to combined VEGF + PDGF for 24 h (Figure 1I) had reduced nuclear factor kappa B p65 translocation. Conditioned medium from macrophages incubated with PDGF and combined VEGF + PDGF in inflammatory conditions promoted phagocytosis by OECs (48 h MCM; Figure 3K). The growth factor treated conditioned media also modulated the expression of genes associated with nerve repair and myelin expression in OECs.

In the context of SCI, macrophages exist throughout all the phases of injury, progressing inflammation or promoting repair, where their actions are modulated by the presence of local environmental cues [8]. These cues can range from cytokines and growth factors to the presence or absence of other cell types. Due to the constant variation of cytokines, the macrophage polarisation shifts between M1 (pro-inflammatory) and M2 (anti-inflammatory) and a dynamic inflammation-healing profile ensues [53]. Additionally, the presence of myelin debris and transplantation of cells can also alter the macrophage polarisation state [54,55,56]. Concurrently, macrophages and inflammatory media can modulate the transplanted cells by altering their secretome profile [57]. OECs have shown promising potential in aiding recovery from traumatic SCI [58,59]. However, few studies exist that explore OEC and macrophage interactions. While it is known that OECs express both macrophage migration inhibitory factor (MIF) and its binding partner [32], it is not yet clear why in some OEC transplantation studies, macrophages tend to populate the centre of the injury site [60] and in others they do not [61]. This difference can only be speculated to be because of the preferential growth of one state (anti-inflammatory, M2) over the other (pro-inflammatory, M1), in the presence of OECs.

Angiogenic growth factors such as VEGF and PDGF have been used as potential candidates for therapy in pre-clinical studies of SCI [29,30]. While individual studies with the growth factors in separate systems have shown that both VEGF and PDGF enhance macrophage recruitment and induce M2 activation status [20,62], combined VEGF + PDGF showed reduced activation [30]. Thus, the effects of these growth factor combinations on macrophage activation with subsequent effects on secretion pattern could be important for regulating the functioning of a safe and favourable cell transplantation therapy.

We first examined how inflammatory conditions affected the translocation of NF-κB p65 subunit from cytoplasm to nucleus in macrophages, which is a crucial mechanism for inducing transcription of pro-inflammatory genes [63]. Although our inflammatory media maintained high translocation at 3 h and 24 h, by 48 h it had reduced. This trend has been previously observed in neutrophils of patients having suffered major trauma where the rate of translocation had reduced by 12 h [64]. This can be attributed to reduction of NF-κB availability over time or reduced receptor activation due to persistence of inflammatory mediators, leading to reduced activity of the inhibitor of kappa kinase (IKK) complex by negative feedback [65].

Interestingly, when VEGF, PDGF and combined VEGF + PDGF were added to macrophages, they greatly influenced NF-κB p65 translocation. At 24 h, combined VEGF + PDGF significantly reduced translocation (Figure 1I). By 48 h, there was a reduction in translocation for all the growth factors except for PDGF (Figure 1L). PDGF maintained higher NF-κB activation along with combined VEGF + PDGF compared to control and VEGF. Previous studies have shown that PDGF, through the Ras/phosphatidylinositol-3-kinase (PI-3-K)/Akt pathway, maintains NF-κB activity [66]. For VEGF, there is a lack of definite studies on NF-κB activation kinetics in macrophages. However, previously, in hematopoietic progenitor cells, it was shown that VEGF inhibits NF-κB activation by VEGFR kinase-independent inhibition of IKK [67]. Thus, this may contribute to the slow activation of VEGF compared to PDGF or combined VEGF + PDGF. Additional involvement of inflammatory mediators appeared to overshadow the effect of individual growth activators during NF-κB activation at 3 (Figure 1G) and 24 h (Figure 1J). Interestingly, at 48 h, the activation status of macrophages was similar to one seen with only growth factor (Figure 1L,M), except that VEGF in inflammatory media was significantly higher in activation compared to inflammatory media alone. Since VEGF had a slow activity in NF-κB activation status, it may be that VEGF had a negligible role in NF-κB translocation at a longer time point (48 h).

Our next aim was to determine the inflammatory cytokine profile of the macrophages at three time points: 3 h, 24 h and 48 h. Consistent with our NF-κB translocation study, we found inflammatory media increased expression of TNF-α gene and cytokines, throughout all the three MCMs (Figure 2A,D). However, IL-6 expression was only evident at 48 h while cytokine levels were observable at both 24 and 48 h (Figure 2). Thus, the gene and cytokine profile of TNF-α and IL-6 varied from each other considerably. While VEGF and PDGF, individually, reduced NF κB p65 translocation at 3 h treatment (Figure 1F), the resultant change in cytokine level of TNF-α was not detectable until 48 h (Figure 2E). Moreover, in trend with our NF κB translocation study at 24 h, combined VEGF + PDGF showed significant reduction (Figure 1I), compared to control and PDGF, and there was significant reduction in IL-6 gene expression when compared to PDGF (Figure 2H). While it is known that VEGF, being an anti-apoptotic mediator, can suppress TNF-α secretion, PDGF is associated with providing protection from TNF-α-induced death by antagonising its activity [68,69]. This was also evident when we introduced PDGF in inflammatory media and observed that it provided protection by suppressing TNF-α expression. Contrary to this, PDGF-BB has also been linked with increased expression of IL-6, which was evident from our PCR finding [70]. Thus, our cytokine data reveals a beneficial role of PDGF in inflammatory media (24 h MCM) by reduction of a major pro-inflammatory cytokine, TNF-α, without any change in IL-6.

To determine the paracrine effects of these mediators released from macrophages on OECs, we assessed their ability to clear myelin debris and the modulation of expression of genes related to nerve repair and myelination. Our findings showed that the OECs in MCM with inflammatory media had a phasic characteristic in their phagocytic ability of myelin debris, i.e., was lower in efficacy in 3 h MCM, then increased in 24 h MCM, and was again lower in 48 h MCM (Figure 3I). Out of all the growth factor combinations, PDGF induced the most efficient phagocytic ability in non-inflammatory unconditioned media and conditioned media (Figure 3). Additionally, PDGF in inflammatory media as well induced efficient phagocytosis in 48 h MCM.

To determine how unconditioned and macrophage conditioned media affected the “nerve repair associated genes” and “myelin expression associated genes”, we assessed the genes responsible for the two effects in OECs. Many of the genes represented in these groups have previously shown positive outcomes in nerve repair [71,72,73,74] or have been detected in different stages of maturation in another closely related glial cell, Schwann cells (SCs), for myelination [75,76,77]. Since both OECs and SCs share a common embryological origin, we wanted to explore the expression of these genes in OECs [78]. Our comparisons showed that unconditioned inflammatory media upregulated all of the nerve repair genes (Figure 4A), while 3 h MCM could only upregulate *Jun* and *Ngfr* while downregulating *Sox2* and *Gdnf*. Additionally, only three myelin expressing genes were upregulated by unconditioned inflammatory media, while inflammatory MCM had a negligible effect (Figure 5). Thus, we could infer that inflammatory media can greatly influence nerve repair genes over myelination in OECs, which is desirable in any transplantation. Amongst the growth factors, PDGF, promoted the best outcomes in nerve repair genes by upregulating four genes in unconditioned media and three genes in conditioned media (Figure 4). However, combined VEGF + PDGF under inflammation in conditioned media promoted better outcome by upregulating *Sox2* and *Gdnf* than either VEGF or PDGF, which only upregulated *Bdnf* and *Sox2*, respectively. Further, for myelin expressing genes, unconditioned media containing growth factors in the presence and absence in inflammatory media had negligible effects which downregulated most of the genes (Figure 5). However, conditions containing PDGF and combined VEGF + PDGF promoted upregulation of most other genes. While PDGF promoted *Mpz*, combined VEGF + PDGF upregulated *Srebf1* and *Pou3f1*. Additionally, VEGF + PDGF in inflammatory conditioned media promoted both *Pou3f1* and *Mpz*, whereas PDGF promoted only the latter. Thus, both PDGF and combined VEGF + PDGF had beneficial effects on OEC gene expression.

Although this study provides a novel interaction study between macrophages and OECs and can be beneficial in the context of SCI therapy, there are certain limitations. In the context of SCI, bone marrow derived macrophages play a major role, and this may have a variable in vitro outcome in comparison to peritoneal macrophages, as used in this study. Thus, further confirmatory tests would be required to validate. While our NF-κB study corroborated with the majority of cytokine findings, further assessment with protein quantification in both cytoplasm and nuclei in macrophages would provide confirmation of our findings. In addition, all our findings are based on 70–80% of the OEC population. Further purification strategies can be employed to look at defined outcomes.

In conclusion, the transplantation of OECs following the administration of growth factors may have beneficial effects for neural repair. To optimise the therapy, the duration of growth factor treatment should be kept in mind, as patterns of OECs’ functional and gene expression vary over time and can be influenced by the paracrine effects of macrophages. Both PDGF and combined VEGF + PDGF showed beneficial effects like reducing pro-inflammatory cytokines (TNF-α) by reduction of NF-κB translocation, promoting phagocytosis of myelin debris and promoting nerve repair genes in OECs either directly or through paracrine effects of macrophages. While in vitro responses may differ from in vivo responses, this work provides a foundation for a potential combinational therapy of VEGF and PDGF with OECs for SCI treatment.

## Figures and Tables

**Figure 1 cells-11-02408-f001:**
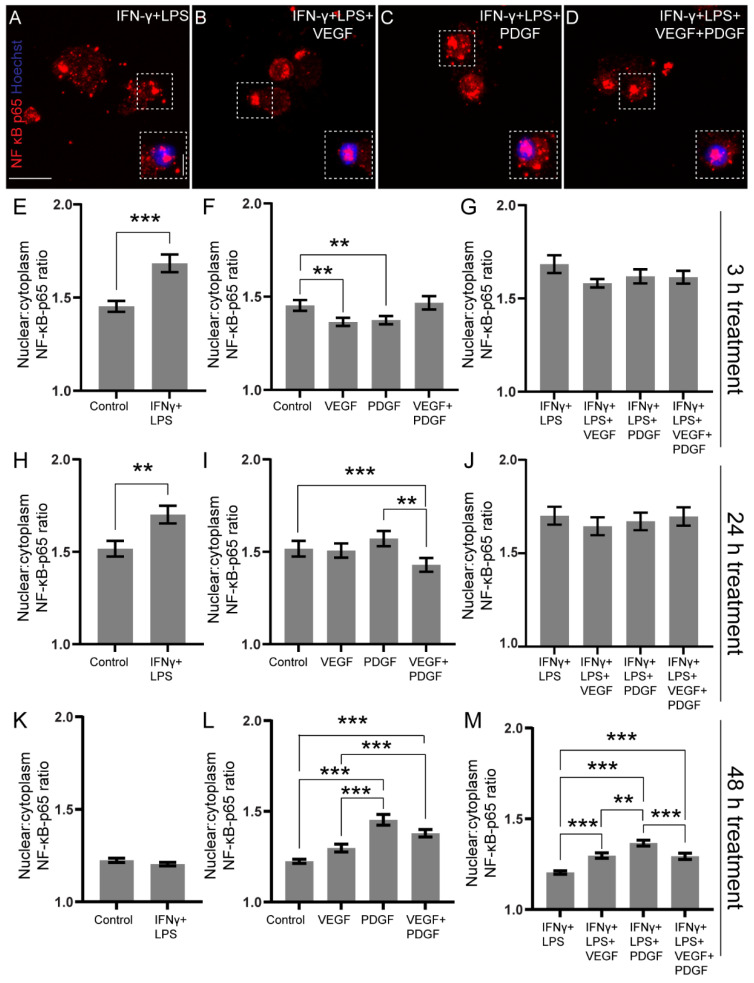
NF-κB p65 translocation in primary macrophages treated with inflammatory, VEGF, PDGF and combined VEGF + PDGF for 3 h (early), 24 h (middle) and 48 h (late). (**A**–**D**) Confocal representative images of primary macrophages showing NF-κB p65 translocation (red) from cytoplasm to nucleus (blue), treated for 48 h with (**A**): inflammatory media (IFN-γ + LPS), (**B**): IFN-γ + LPS + VEGF, (**C**): IFN-γ + LPS + PDGF, and (**D**) IFN-γ + LPS + VEGF + PDGF. Scale: 15 µm (5 µm for inset) across all images. Graphs represent NF-κB p65 nuclear: cytoplasm intensity ratio (a measurement of activation status of macrophage by nuclear translocation) when treated for (**E**–**G**) 3 h with (**E**) control and IFN-γ + LPS, (**F**) control, VEGF, PDGF and combined VEGF + PDGF, (**G**) VEGF, PDGF and combined VEGF + PDGF in inflammatory media (IFN-γ + LPS); (**H**–**J**) 24 h with (**H**) control and IFN-γ + LPS, (**I**) control, VEGF, PDGF and combined VEGF + PDGF, (**J**) VEGF, PDGF and combined VEGF + PDGF in inflammatory media (IFN-γ + LPS) and (**K**–**M**) 48 h with (**K**) control and IFN-γ + LPS, (**L**) control, VEGF, PDGF and combined VEGF + PDGF, (**M**) VEGF, PDGF and combined VEGF + PDGF in inflammatory media (IFN-γ + LPS). ** *p* ≤ 0.01, *** *p* ≤ 0.001 measured using unpaired *t*-test with Welch’s correction for (**E**,**H**,**K**) and one-way ANOVA followed by Dunn’s multiple comparisons test for (**F**,**G**,**I**,**J**,**L**,**M**). Error bar represents mean ± SEM for three technical replicates of three biological replicates.

**Figure 2 cells-11-02408-f002:**
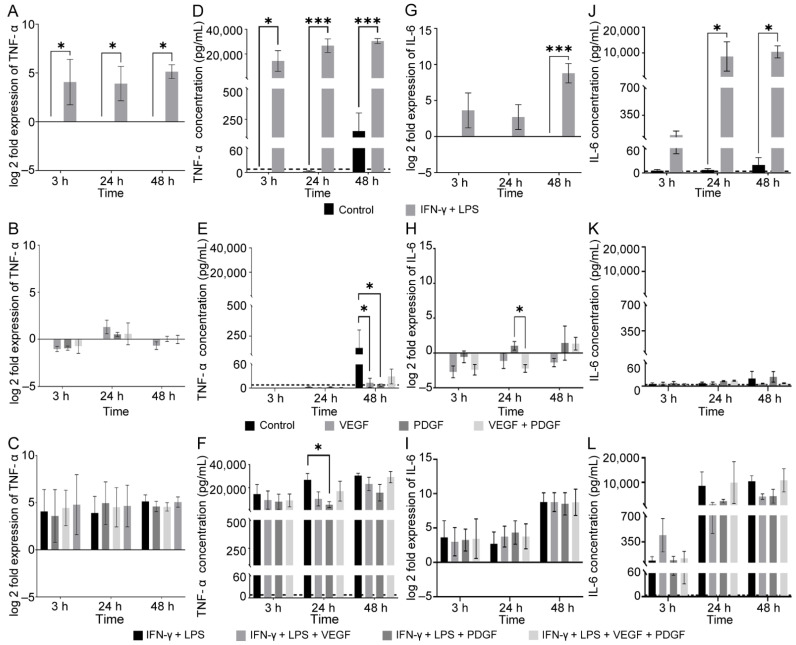
Effects of VEGF and PDGF on TNF-α and IL-6 gene expression and secretion by primary macrophages in non-inflammatory and inflammatory conditions. Graphs represent both qPCR and ELISA results of macrophage pro-inflammatory cytokine profiles when treated with VEGF, PDGF (single or combined) in the presence or absence of inflammatory medium for 3, 24 and 48 h. (**A**–**D**) show control versus IFN-γ + LPS (**A**–**F**) shows TNF-α gene expression for (**A**) control vs. IFN-γ + LPS, (**B**) control, VEGF, PDGF and VEGF + PDGF, (**C**) VEGF, PDGF and VEGF + PDGF in inflammatory media (IFN-γ + LPS); (**D**–**F**) TNF-α cytokine levels (**D**) for (**A**), (**E**) for (**B**) and (**F**) for (**C**). (**G**–**L**) shows IL-6 gene expression for (**G**) control vs. IFN-γ + LPS, (**H**) control, VEGF, PDGF and combined VEGF + PDGF, (**I**) VEGF, PDGF and combined VEGF + PDGF in inflammatory media (IFN-γ + LPS); (**J**–**L**) IL-6 cytokine levels (**J**) for (**G**), (**K**) for (**H**) and (**L**) for (**I**). Dashed lines for each cytokine represent lowest detectable levels for the kit: TNF-α: 8 pg/mL and IL-6: 4 pg/mL. * *p* ≤ 0.05, *** *p* ≤ 0.001, measured using two-way ANOVA with post-hoc Fisher’s LSD multiple comparisons test. Error bar represents mean ± SEM for three technical replicates of three biological replicates.

**Figure 3 cells-11-02408-f003:**
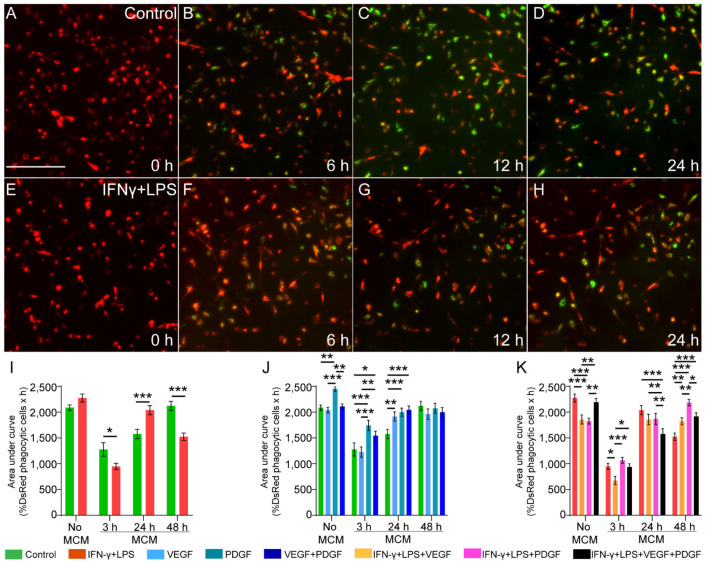
Effect of unconditioned and macrophage conditioned media (MCM) with or without VEGF, PDGF on phagocytosis of myelin debris by primary OECs. (**A**–**H**) Time lapse images of DsRed OECs (red) phagocytosing myelin debris (green) at 0, 6, 12 and 24 h. Shown are representative time-lapse images for OECs in the (**A**–**D**) absence and (**E**,**F**) presence of 3 h pro-inflammatory MCM. Scale bar: 200 μm. (**I**–**K**) Graphs show area under curve (AUC) for phagocytosis of myelin debris with (3, 24, 48 h treated) or without MCM for (**I**) control and IFNγ + LPS, (**J**) control, VEGF, PDGF and combined VEGF + PDGF, (**K**) VEGF, PDGF and combined VEGF + PDGF in inflammatory media (IFNγ + LPS)**.** * *p* ≤ 0.05, ** *p* ≤ 0.01, *** *p* ≤ 0.001 measured using unpaired *t*-test for (**A**); two-way ANOVA with Fisher’s LSD test for (**B**,**C**). Error bar represents mean ± SEM for three technical replicates of three biological replicates.

**Figure 4 cells-11-02408-f004:**
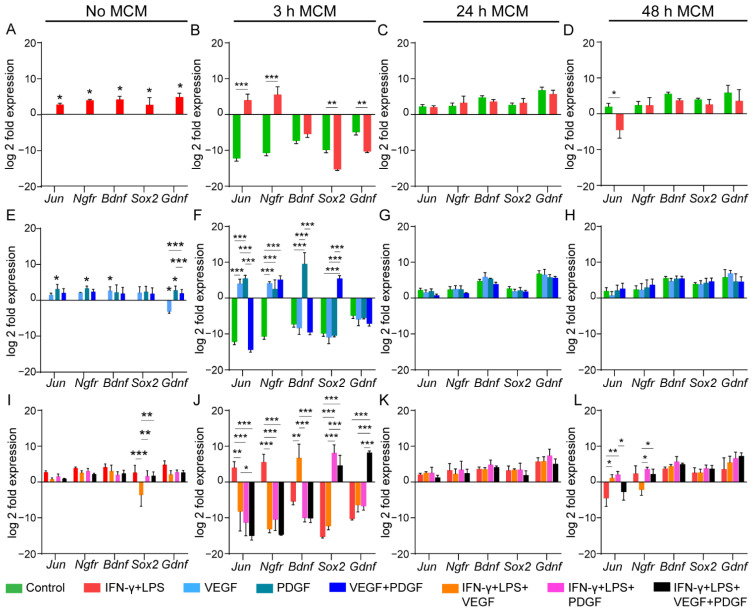
Effect of unconditioned and macrophage conditioned media (MCM) with or without VEGF, PDGF on nerve repair associated gene expression in primary OECs. (**A**–**L**) Graphs represent the log 2-fold expression of genes responsible for nerve repair in OECs when treated with unconditioned media (No MCM), 3 h, 24 h and 48 h MCM. (**A**–**D**) Untreated control and IFN-γ + LPS media for (**A**) Unconditioned media, (**B**) 3 h MCM, (**C**) 24 h MCM, (**D**) 48 h MCM; (**E**–**H**) Untreated control vs. VEGF vs. PDGF vs. combined VEGF + PDGF (**E**) Unconditioned media, (**F**) 3 h MCM, (**G**) 24 h MCM, (**H**) 48 h MCM; (**I**–**L**) IFN-γ + LPS vs. IFN-γ + LPS + VEGF vs. IFN-γ + LPS + PDGF vs. IFN-γ + LPS + VEGF + PDGF for (**I**) Unconditioned media, (**J**) 3 h MCM, (**K**) 24 h MCM, (**L**) 48 h MCM. Normalisation was done on no MCM control data. * *p* ≤ 0.05, ** *p* ≤ 0.01, *** *p* ≤ 0.001 measured using unpaired *t*-test for (**A**–**D**); two-way ANOVA with Fisher’s LSD test for (**E**–**L**). Error bar represents mean ± SEM for three technical replicates of three biological replicates.

**Figure 5 cells-11-02408-f005:**
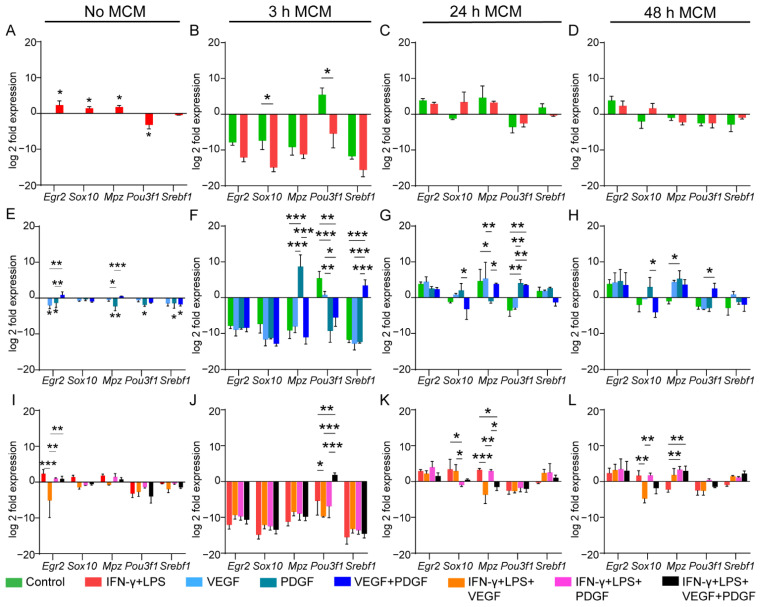
Effect of unconditioned and macrophage conditioned media (MCM) with or without VEGF, PDGF on myelination associated gene expression in primary OECs. (**A**–**L**) Graphs represent the log 2-fold expression of genes responsible for myelination in OECs when treated with unconditioned media (No MCM), 3 h, 24 h and 48 h MCM. (**A**–**D**) Untreated control and IFN-γ + LPS media for (**A**) Unconditioned media, (**B**) 3 h MCM, (**C**) 24 h MCM, (**D**) 48 h MCM**;** (**E**–**H**) Untreated control vs. VEGF vs. PDGF vs. combined VEGF + PDGF (**E**) Unconditioned media, (**F**) 3 h MCM, (**G**) 24 h MCM, (**H**) 48 h MCM; (**I**–**L**) IFN-γ + LPS vs. IFN-γ + LPS + VEGF vs. IFN-γ + LPS + PDGF vs. IFN-γ + LPS + VEGF + PDGF for (**I**) Unconditioned media, (**J**) 3 h MCM, (**K**) 24 h MCM, (**L**) 48 h MCM. Normalisation was done on no MCM control data. * *p* ≤ 0.05, ** *p* ≤ 0.01, *** *p* ≤ 0.001 measured using unpaired *t*-test for (**A**–**D**); two-way ANOVA with Fisher’s LSD test for (**E**–**L**). Error bar represents mean ± SEM for three technical replicates of three biological replicates.

## Data Availability

Data can be provided upon reasonable request to the corresponding author.

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
