# Peer review of "Macrophages Treated with VEGF and PDGF Exert Paracrine Effects on Olfactory Ensheathing Cell Function"

_cells, 2022, doi:10.3390/cells11152408_

Round 1

Reviewer 1 Report

The authors present an in vitro approach to dissect the complex interplay of environment and signalling influence on cellular function in SCI. In particular they investigate the combinatorial roles VEGF and PDGF signalling molecules and inflammatory triggers to influence the the ability of the Macrophage population to instruct OEC function via paracrine signalling. The experiments are (mostly) well controlled, well powered, and subjected to statistical analysis. The figures are nicely assembled. The introduction was well written.

The authors first study the effect of VEGF/PDGF/LPS+INFg on the inflammation status of macrophages. They use 3 hallmark markers of inflammation (NFKB nuclear translocation, TNFa and IL6 expression). Whilst differences can be identified in certain combinations, at certain time points, there was no consistent outcome; and some conflicting outcomes, suggesting complexities exist beyond the scope of the experimentation. These should be acknowledge and discussed in better detail. The authors then use a clever live cell myelin phagocytic assay to assess OEC function. Differences are identified and dependent on the combination of factors assayed. Lastly, the authors use qPCR for a set of genes proposed to inform on the nerve repairing and myelination function of OECs. Once again some differences are identified.

Unfortunately, the challenge of how to interpret these differences in a biological sense, or with a targeted purpose of informing on rational design of use of OECs in SCI remains an open question. The abstract does describe some key ‘take home messages’ but these are difficult to pick from the paper amongst all of the other data presented. The results and discussion describes the graphs, and data at great depth, but does little to enlighten on this big picture. This is highlighted by the summary of the paper, which (to paraphrase) says treatments ‘may have effects’ differences need to be ‘kept in mind’. The discussion missed this opportunity synthesize the data into actionable outcomes that inform on the rational use of OECs in SCI.. There was also little cross-examination/interpretation of the different assays, which could provide insight into the predicted functional axis of NFKB translocation/inflammatory cytokine production / impact on phagocytic and nerve repair function.

The results section was densely written, and hard to absorb and read. It took a lot of effort to comprehend. The results text lacked a narrative of what it was trying to be achieve through all the different conditions - why different time points are important, why differences between time points are important. On top of this, there were misleading, vague or long and convoluted sentences. The authors have attempted to describe almost every difference in the data but only minor insights as to the purpose of experiments, what outcomes are expected and they inform on the next line of assay. This made the section very ‘dry’ and hard to follow.

Altogether, there is a wealth of data provided, but it is description, and discussion, and focused interpretation should be improved. I am unclear about the major findings as a result. I recommend revising to ensure clean and easy messaging (rather than swamping with data), better narrative of the experimental reasoning (potentially underpinned by hypothesis also), and converging different data sets into major outcomes that help understand the rationale use and priming of OECs for treatment of SCI.

Some more additional detailed comments listed below.

Intro.

Page 2 Line 74: “….. they lacked an in vitro understanding of the findings which would have given us a better understanding of the perturbations in the microenvironment.” It is not obvious why ‘in-vitro’ understanding is critical. It seems what is lacking is an understanding of how different cell populations in the SCI are collaborating/interacting to facilitate the effect. An in-vitro in –vitro model can be used to help dissect these, but the ultimate goal is to impose such findings to understand in-vivo application.

Figure 1 and 2. Authors should comment on the divergent differences observed using different markers of inflammation (NFKB vs TNFa vs IL6) equivalent time’s points / treatment conditions. One may have predicted these reporters of inflammation to converge in response. @ Line 336: It would be good to summarise here the take home message by aggregating the information from these three markers as it is otherwise difficult to make sense of all the different data points as a collective.

All figures. What is the reasoning behind 3, 24 and 48 hour MCM conditions. How does this relate to the in-vivo environment, or inform use of OEC in SCI.

Line 258. “As survival and function of OECs that are transplanted into the inflammatory injury site is influenced by the resident cells including macrophages, and these responses can be modulated by the application of the growth factors VEGF and PDGF, we needed to assess how these various factors interact.”. This statement provides critical precedence to the current study but has no references.

Line 373. It is critical to first address the effect of macrophage conditioned media on OEC function I see the data in Supp figure, but it is not described in the text until line 437 – at the end of the section. This seems a critical piece of information to describe first. Along the same lines, describe how OECs respond to un-conditioned media in the presence of LPS+INFg, VEGF and PDGF. These data are required to dissect the individual contributions of LPS, macrophage paracrine factors, and added growth factors in the assays presented in Fig 3.

Figure 3: What is the significance of the time course 0-24 hours in phagocytosis assays Fig 3? It is great to have all the data, but the kinetics of the uptake seems superfluous to the overall impact this would have in a SCI model. With that said, I like the AUC analysis, and feel it worth considering to use this as the main data, rather than sup data.

Line 391. The comparisons described in this sentence are vague. Revise.

Line 407: (Figure L)

Line 421. This sentence is confusing and the figure is not referenced – is it data from 3I vs 3O?

Line 446. This sentence fails to capture the overall positive or negative effects. It needs revision. All up, this section is overload with so many comparisons it is hard to make easy sense of it – cant see the forest for the trees. The summary here is thus most important. If I am correct, the data suggest that a 3h treatment of OEC with MCM inhibits their phagocytic capacity and in general, this effect is alleviated with longer MCM treatments, and in MCM created through supplementation of PDGF and combined PDGF + VEGF.

Line 470-471: Add some reference and details explaining how the genes listed contribute to the stated functions in OECs.

Line 481: Word processing error messaging is present in this section.

Line 475: It is stated the OECs are cultured with macrophages –this is incorrect I believe.

Line 478: It is not clear what all the data described in the entire sections related to Figure 5 and 6 is compared against. For example, Figure 5A, control conditions, genes are up or downregulated, relative to what though? It is the same for every other panel in figures 5 and 6.

Line 482 what does “When we compared the basal effects of growth factor treated MCM” mean?

Line 488 “Further, when we challenged the growth factors with inflammatory MCM, inflammatory only media upregulated Jun and Ngfr expression compared to all other growth factor combinations.” Confusing.

It is challenging to easily follow the descriptions in much of this section, and so suggest major revision. Throughout the blocks of associated text, it is confusing (1) when controls are described as being up and down regulated (2) too many combinations and permutations of comparisons being made (and the vague descriptions of them) and (3) leading to a lack sense what it all means, what is important, what is not. After reading several times it remained hard to understand, and pin-point major findings. It needs revision. One suggestion would be to have some hypothesis stated upfront and describe within that ‘more focussed’ objective, rather than describing everything. As per last section - cant see the forest for the trees. In the end, the depth, complexity, and convoluted nature these decriptions is out of balance with the final summary of this data at line 584 “Thus, our PCR data showed that there was a variable difference in expression patterns of genes in OECs when treated with growth factors in the presence and absence of 585 inflammatory mediator either directly or through MCMs”. The data needs be vastly summarised, possibly supplementary, as it was very challenging (and even distracting given the summary).

Discussion.

In general, there is trouble finding the exact data from which conclusions in discussions are drwn, it is not obvious. Reference to the data, including panels (given data is variable across time and conditions)

Line 592: Macrophages exposed to combined VEGF + PDGF had reduced nuclear factor kappa B p65 translocation…..”. It appears speculative to arrive at this conclusion based on all data in figure 1.

Line 593 “Conditioned medium from macrophages incubated with PDGF and combined VEGF + PDGF in inflammatory conditions promoted phagocytosis by OECs”. Please explicitly clarify what conditions you refer to as the sentence be interpreted in different ways. Refer to the figure panels where this data is derived from, it is not onbvious when looking at the entire figure.

Line 601 “Due to constant variation of cytokines, the macrophage polarisation shifts and a dynamic inflammation-healing profile ensues “.  What is the shift? What does inflammation-healing mean ?

Line 609 “it is not yet clear why in some studies macrophages tend to populate the centre of injury site [56] and in others they do not [57]” For clarity, state that you refer to OEC transplantations studies.

Line 628 “Interestingly, when VEGF, PDGF and combined VEGF + PDGF were added to macrophages, the individual factors started to translocate albeit slowly.” Vague, be explicit.

Line 630: The term ’rate’ is misleading given no rate (translocation/time) data is synthesized per se.

Line 630 “the reduction of translocation rate was even slower, especially for PDGF” is a confusing statement. Rephrase. The next sentence is also confusing. Reference the data may help, but a re-write of lines line 628-632.

Line 644. Please add the take home message. I struggle to understand what the conclusions of this body of data means, what the time course is about, how this informs on macrophage function in SCI.

Line 645 “Our next aim was to determine the inflammatory cytokine profile of the macrophages at three time points following the NF-κB translocation” is misleading as the time points are not in relation to that of when NFKb translocates.

Line 648 “At longer time points, both VEGF and PDGF reduced the expression of TNF-α”. Longer? 48 hour time point.

Line 656 “Based on this inflammatory profile of macrophages” No context as to how these markers are interpreted is given. It is not clear how the data described above is consistent with this conclusion. If anything, reduced expression IL6 / TNFa suggests non-inflammatory profile.

Line 661 “Out of all …….” Sentence is long and confusing. It is not clear which ‘times points’ are being referenced (3, 24, 48h and/or the time course of the assay).

Line 666” we next assessed the genes responsible for the two effects as seen in another glial cell, Schwann cells [33]”. This reference does not show that these genes are responsible for the functions. Please add the correct references.

Line 670 “This led us to speculate that earlier time points of inflammatory macrophage can induce neuroprotective OECs compared to myelinating OECs.”. Add more detail, it is not obvious how the genes report on “neuorprotective” – this was introduced – do you mean nerve-repairing, myelination or something else. That some of genes are up, and some or down suggest this suite of genes is insufficient/underpowered to report on such functions.

Line 692-698. The conclusion is passive. Reading the whole paper, one would expect that some positive conclusion can be drawn from all the data (e.g. as stated in the abstract). I agree that difference are identified, but synthesis of the data into a meaningful biological outcome is lacking.

The limitations to the study should be acknowledged and discussed where appropriate.

Reviewer 2 Report

I am writing about the manuscript entitle: Macrophages treated with VEGF and PDGF exert paracrine effects on olfactory ensheathing cell function.

In the present in vitro study the authors have demonstrated the effect of VEGF and PDGF on macrophages in an inflammatory condition, suggesting that the use of these growth factors together with OEC transplantation may have beneficial effects for neural repair. 

The manuscript is well-designed and well-written; Materials and Methods are clear and organized and the figures are very nice.

In my opinion the ms can be accepted in present form for publication.

Author Response

The reviewer made no suggestions for changes and recommended the paper be accepted in the original format. However, as per the suggestions from Reviewers 1 and 3, we have simplified the description of the Results and the Discussion

Reviewer 3 Report

The manuscript entitled "Macrophages treated with VEGF and PDGF exert paracrine effects on olfactory ensheathing cell function" by Basu, et al. focuses on investigating immunomodulatory effects of conditioned macrophage on the functionality of OECs in vitro. This is an interesting work, which provides important evidences on further application of cell-based therapy for SCI. 

Major comments:

1.     Figure 1: There are only confocal images on NF-κB p65 translocation in the cells with and without inflammatory condition. It is better to include images in the cells under inflammatory condition with VEGF, PDGF or combined VEGF + PDGF treatment.

2.     Figure 3: Similarly, there are only images on myelin debris with and without inflammatory condition. It is better to include images under inflammatory conditions with VEGF, PDGF or combined VEGF + PDGF treatment.

3.     As the studies are related to different types of condition, the description of results can be further improved with simplicity and clarity.

Round 2

Reviewer 3 Report

The authors have addressed all comments. I have no further comments.